# Glucose Homeostasis, Metabolomics, and Pregnancy Outcomes After Bariatric Surgery (GLORIA): Protocol for a Multicentre Prospective Cohort Study

**DOI:** 10.3390/jcm14134782

**Published:** 2025-07-07

**Authors:** Ellen Deleus, Niels Bochanen, Dries Ceulemans, Hanne Debunne, Bénédicte Denys, Roland Devlieger, Ina Geerts, Annouschka Laenen, Lisbeth Jochems, Els Lannoey, Matthias Lannoo, Anne Loccufier, Toon Maes, Joke Marlier, Astrid Morrens, Nele Myngheer, Luna Tierens, Griet Vandenberghe, Annick Van den Bruel, Lien Van den Haute, Bart Van der Schueren, Inge Van Pottelbergh, Katrien Benhalima

**Affiliations:** 1Clinical and Experimental Endocrinology, Department of Chronic Diseases and Metabolism, KU Leuven, 3000 Leuven, Belgiumkatrien.benhalima@uzleuven.be (K.B.); 2Department of Abdominal Surgery, University Hospitals Leuven, 3000 Leuven, Belgium; 3Department of Endocrinology, Diabetology & Metabolism, Antwerp University Hospital, 2650 Edegem, Belgium; 4Department of Development and Regeneration, KU Leuven, 3000 Leuven, Belgium; 5Department of Obstetrics and Gynaecology, University Hospitals Leuven, 3000 Leuven, Belgium; 6Department of Obstetrics and Gynaecology, ZAS Palfijn, 2170 Merksem, Belgium; 7Department of Obstetrics and Gynaecology, AZ Groeninge Kortrijk, 8500 Kortrijk, Belgium; 8Center of Biostatics and Statistical Bioinformatics, KU Leuven, 3000 Leuven, Belgium; 9Department of Obstetrics and Gynaecology, Antwerp University Hospital, 2650 Edegem, Belgium; 10Department of Obstetrics and Gynaecology, Imelda Bonheiden, 2820 Bonheiden, Belgium; 11Department of Obstetrics and Gynaecology, AZ Sint-Jan Brugge, 8000 Brugge, Belgium; 12Department of Endocrinology, Imelda Hospital Bonheiden, 2820 Bonheiden, Belgium; 13Department of Endocrinology, Ghent University Hospital, 9000 Gent, Belgium; 14Department of Endocrinology, ZAS Palfijn, 2170 Merksem, Belgium; 15Department of Endocrinology, General Hospital Groeninge Kortrijk, 8500 Kortrijk, Belgium; 16Department of Obstetrics and Gynaecology, Ghent University Hospital, 9000 Gent, Belgium; 17Department of Endocrinology, AZ Sint-Jan Brugge, 8000 Brugge, Belgium; 18Department of Obstetrics and Gynaecology, AZORG, 9300 Aalst, Belgium; 19Department of Endocrinology, University Hospitals Leuven, 3000 Leuven, Belgium; 20Department of Endocrinology, AZORG, 9300 Aalst, Belgium

**Keywords:** Roux-en-Y gastric bypass, sleeve gastrectomy, continuous glucose monitoring, post bariatric hypoglycaemia, pregnancy outcome, small for gestational age, premature birth, foetal growth restriction

## Abstract

**Background:** Metabolic bariatric surgery is a highly effective and long-lasting treatment for obesity and related chronic conditions. Women of reproductive age represent the largest group undergoing these procedures. Observational studies suggest an increased risk of preterm birth and impaired foetal growth in this population, though the underlying mechanisms remain unclear. A key hypothesis is that altered glucose metabolism, characterised by frequent hypoglycaemia and glycaemic fluctuations, may contribute to these adverse outcomes. While glycaemic variability following metabolic bariatric surgery has been documented, its pattern during pregnancy and impact on pregnancy outcomes are still underexplored. **Methods:** In this Belgian multicentre prospective cohort study, we will investigate glycaemic patterns during pregnancy in women who have undergone metabolic bariatric surgery. Women aged 18–45 years with a confirmed singleton pregnancy up to 11 weeks and 6 days and a history of Roux-en-Y gastric bypass or sleeve gastrectomy will be eligible for inclusion. Women with pregestational diabetes or those taking medication known to interfere with glucose metabolism will be excluded. All participants will receive blinded continuous glucose monitoring (Dexcom^®^ G6) for a 10-day period at four time points throughout the pregnancy. Foetal body composition and growth will be measured during routine ultrasound; skinfolds will be measured in the neonate. The primary outcome is the association between mean glycemia and glycaemic variability on continuous glucose monitoring and birth weight. The planned sample size is ninety-five women. Linear mixed models for repeated measurements will be used for analysis. Confounders such as smoking, micronutrient deficiency, and surgery-to-conception interval will be added to the model as covariates. In a second exploratory phase, each participant in the surgical group will be matched with a control participant—without a history of metabolic bariatric surgery—based on pre-pregnancy BMI and age. Control participants will undergo the same study procedures, allowing for exploratory comparison of glycaemic patterns and other study outcomes. **Discussion:** This prospective longitudinal study will be the largest study using continuous glucose monitoring to investigate glucose metabolism during pregnancy after metabolic bariatric surgery and its impact on foetal growth and newborn body composition. **Trial registration**: ClinicalTrials.gov: NCT05084339. Registration date: 15 October 2021.

## 1. Background

Obesity is associated with increased risks for miscarriages, gestational diabetes mellitus (GDM), preeclampsia, and caesarean delivery [1]. Metabolic and bariatric surgery (MBS) is the most effective treatment for obesity and its related comorbidities [2]. Of all patients who undergo MBS, up to 80% are women of childbearing age [3]. As such, pregnancies in women with a history of MBS are becoming increasingly prevalent. Surgically induced metabolic changes benefit mother and child but can also lead to adverse pregnancy outcomes. Several nationwide register-based cohort studies and systematic reviews have shown that MBS is associated with lower risks for macrosomia, GDM, and hypertensive disorders, but with increased rates of small-for-gestational-age (SGA) infants, preterm delivery, and perinatal mortality [4,5,6,7,8]. Children born SGA are at an increased risk of developing type 2 diabetes and cardiovascular diseases in the long-term [9]. The fact that the increased risks of SGA and preterm deliveries remain when matching for pre-pregnancy weight suggests that MBS as such is associated with adverse obstetric outcomes, and not only through weight reduction [7]. A continuous decline in foetal growth throughout pregnancy has been demonstrated after Roux-en-Y gastric bypass (RYGB), starting at the beginning of the second trimester [10]. In addition, the foetal abdominal subcutaneous tissue thickness (FASTT) was markedly decreased in foetuses of mothers with a history of MBS [11].

The underlying mechanisms of foetal growth restriction in this population remain unclear but are likely multifactorial. Potential contributors include nutrient deficiencies, placental dysfunction, altered glucose metabolism, and potentially changes in other metabolic pathways. A prospective cohort study including 54 women with previous MBS confirmed reduced foetal growth in pregnancies following surgery and evaluated fetoplacental Dopplers as a proxy for placental function [12]. However, no alterations in fetoplacental circulation or correlations with foetal size were observed [12].

Evidence increasingly indicates that maternal metabolites beyond glucose contribute to foetal growth and newborn adiposity [13]. Metabolomics, the comprehensive analysis of metabolites in biological samples, enables the identification of metabolic biomarkers and the elucidation of mechanisms underlying early-life risk for conditions such as SGA, obesity, and T2DM [13]. The maternal metabolome undergoes temporal changes during pregnancy, resulting in stage-specific metabolite profiles [14]. Recent evidence suggests that maternal concentrations of amino acids, acylcarnitines, lipids, and fatty acids and their metabolites are associated with foetal growth and adiposity, independent of maternal BMI (body mass index) and glycaemia [13]. A systematic review assessing metabolomic predictors of SGA identified fatty acids, phosphosphingolipids, and amino acids as the most commonly reported predictive subclasses [15]. However, studies investigating metabolomic profiles in relation to foetal growth among pregnant women following bariatric surgery remain limited. One small study (25 RYBG and 8 SG) reported significantly lower third trimester serum concentrations of branched-chain and aromatic amino acids, as well as unsaturated lipids in the RYGB group [16]. These metabolite reductions were inversely associated with offspring birthweight [16].

The Pedersen’s hypothesis traditionally links maternal hyperglycaemia to increased foetal growth via foetal hyperinsulinemia—a concept confirmed by the Hyperglycaemia and Adverse Pregnancy Outcomes (HAPO) study, which demonstrated a continuous association between maternal glucose levels and increased birth weight [17]. Conversely, maternal hypoglycaemia—as observed in altered glucose dynamics following MBS—may contribute to foetal growth restriction by limiting glucose availability [18]. A retrospective study in a non-bariatric surgery population confirmed a significant association between hypoglycaemia during OGTT and low birth weight [19]. Similarly, in the MBS population, a small study has shown that women with reactive hypoglycaemia on OGTT had significantly higher rates of SGA infants [20]. A prospective cohort study investigated glucose metrics in 23 pregnant women with RYGB and 23 BMI- and parity-matched pregnant controls using continuous glucose monitoring (CGM) during the first, second, and third trimester, as well as early postpartum [21]. The study found a slight increase in time spent in the low glycaemic range among pregnancies resulting in SGA infants. However, the study was not sufficiently powered to detect significant differences in this outcome.

The main aim of this multicentric prospective study is to determine whether altered glucose metabolism (including lower blood glucose levels, increased time spent in hypoglycaemia, and glycaemic variability) measured by CGM in pregnancy after MBS is related to impaired foetal growth.

## 2. Methods

### 2.1. Study Design and Setting

The GLORIA study is a prospective cohort study involving eight Belgian sites, investigating pregnancy outcomes following sleeve gastrectomy (SG) or RYGB. In a second stage, outcomes will be compared to a matched control group using an exploratory approach. SG has increasingly become the first-choice type of MBS for women of reproductive age. We estimate that between 30 and 50% of participants will have SG in the study. For each participant in the surgical cohort, an age- (≤2 years difference, and age > 40 will be rounded to 40 years) and BMI-matched (≤2 BMI points difference, matched to pre-pregnancy BMI) pregnant woman without history of MBS will be recruited. Seven Belgian hospitals were originally included to take part in the study: three university hospitals (UZ Leuven, UZ Antwerpen, and UZ Gent) and four large non-university hospitals (ZAS Palfijn, AZORG Aalst, Imelda Bonheiden, and AZ St-Jan Brugge). In 2023, one additional centre was added. All eight centres have previously participated in multicentre studies initiated by our research group [22,23]. Participating centres were selected for their expertise in endocrinology research, as well as obstetrical management of pregnant women with a history of MBS.

### 2.2. Modifications to the Study Protocol

One more study site was added to participate in the study (AZ Groeninge Kortrijk) to facilitate recruitment. This has led to an amendment which was approved by the central ethics committee on 17 August 2023.

### 2.3. Objectives

This trial is designed to show with 80% power a linear association between mean glycemia or glycaemic variability and birth weight in a pregnancy after MBS. The hypothesis is that an altered metabolic profile including an altered glucose metabolism (with increased time spent in hypoglycaemia and greater glycaemic variability) and altered metabolomics during pregnancy after MBS contribute to the increased risk of adverse pregnancy outcomes such as SGA and preterm delivery. In addition, we hypothesise that concentrations of branched-chain and aromatic amino acids, and unsaturated lipids in the third trimester of pregnancy, are lower in pregnant women with MBS, and that this is also associated with a lower birth weight.

This study aims to evaluate whether biomarkers derived from CGM and metabolomic profiling in pregnancies following MBS are associated with foetal growth and newborn body composition.

The following specific aims will be addressed.

Primary aim: To determine whether altered glucose metabolism (including lower mean blood glucose levels, hypoglycaemia, and glycaemic variability) measured by CGM in pregnancy after MBS is related to delayed foetal growth.

Exploratory objectives:To compare glucose metabolisms between pregnant women after MBS and age- and BMI-matched pregnant women without MBS.To determine whether metabolic alterations are associated with delayed foetal growth in pregnant women after MBS compared to pregnant women without MBS, and, in particular, which metabolites are related to adverse outcomes in pregnancy.To determine whether there are differences in glucose metabolism and other metabolites (by using metabolomics) between RYBG and SG.To determine the impact of MBS on the body composition of newborns.To determine the accuracy of CGM compared to the self-monitoring of capillary blood glucose (SMBG) to screen for GDM in pregnant women with MBS.

#### 2.3.1. Primary Outcome

In the surgical group, we will assess mean glycaemia as a continuous outcome variable, glycaemic variability (standard deviation) as a continuous outcome variable, and birth weight as a continuous explanatory variable. Linear mixed models will be used for the primary outcome analysis. The outcome variables will be measured at four time points for each subject. Confounders will be added to the model as covariates.

#### 2.3.2. Secondary Outcomes

In the surgical group, we will assess associations between glycaemic patterns and the risk of adverse pregnancy outcomes, including SGA and birth defects. Linear mixed models will be used analogously to the primary analysis, with adverse pregnancy outcomes as explanatory variables in the models. Additional analyses will assess the association of hypoglycaemia as a binary variable with birth weight or adverse pregnancy outcomes. generalised linear mixed models for binary outcomes will be used for data analysis.

#### 2.3.3. Exploratory Outcomes

An overview of clinical definitions and diagnostic criteria considered in this study is given in a Appendix A. The self-designed questionnaire on the user-friendliness of the masked CGM for both groups is given in Appendix A. These outcomes will be analysed for both the surgical and the control group.

-Maternal glycaemic outcomes: percentage of time < 54 mg/dL (ADA recommendations of the definition of hypoglycaemia) [24]; glycaemic variability metrics: standard deviation (SD), Coefficient of Variation (CV) and Mean Amplitude of Glucose Excursions (MAGE); percentage of time > 120 mg/dL; percentage of time > 140 mg/dL; percentage of time > 180 mg/dL; percentage of time < 70 mg/dL; percentage of time < 63 mg/dL; percentage of time < 50 mg/dL; low blood glucose index (LBGI)-Maternal pregnancy outcomes: duration of pregnancy; time from surgery to conception; prevalence of micro- and macronutrient deficiencies; body composition in early pregnancy as assessed by bio-impedance measurement, prevalence of GDM; gestational weight gain (GWG); prevalence of caesarean section; prevalence of pre-eclampsia and eclampsia, prevalence of gestational hypertension; surgery-related complications during pregnancy; type of labour; type of delivery; complications during delivery-Foetal growth and body composition outcomes: prevalence of SGA; large for gestational age (LGA); prevalence of intra-uterine growth retardation (IUGR); FASTT; pregnancy loss-Neonatal outcomes: body composition of the newborn assessed by skin fold thickness measurements, according to [25]; prevalence of preterm delivery; prevalence of neonatal hypoglycaemia; prevalence of neonatal intensive care unit (NICU) admission; foetal malformation; complications during delivery; sex of the infant; birth weight, length, and percentile; neonatal death-Identification of new biomarkers: metabolomics on maternal plasma collected at 30–34 weeks: lipid omics platform and metabolism and energy platform. We will prioritise analyses for the surgical group.-Patient-reported outcomes: certain potential confounders, such as smoking status, will be derived from survey data. Additionally, other patient-reported outcomes collected through the surveys will be analysed as independent outcomes in the exploratory comparisons between subgroups. The following questionnaires will be completed at the different study visits:A self-designed questionnaire on general habits and socio-economic background previously used in the Belgian Diabetes in Pregnancy study (BEDIP) to extensively collect information on socio-economic status and habits [22].A self-designed questionnaire on the user-friendliness of the masked CGM, based on the CGM Satisfaction Scale (CGM-SAT) [26], the Glucose Monitoring Experiences Questionnaire (GME-Q) [27], and the Glucose Monitoring Satisfaction Survey (GMSS) [28].Frequency Food questionnaire (FFQ), including questions on frequency and portion size of consumed foods and beverages validated for the Belgian population [29].International Physical Activity Questionnaire (IPAQ, full version) validated for use in the Belgian population and as used in the BEDIP study [22]. This questionnaire assesses various domains of physical activity, including job-related tasks, caregiving, and time spent sitting.The 36-Item Short Form Health Survey (SF-36) includes a set of generic, coherent, and easily administered quality of life measures and is validated for use in the maternity context [30].The 20-item Centre for Epidemiologic Studies-Depression (CES-D) questionnaire, which is validated in pregnancy to asses symptoms of clinical depression over the past seven days [31].The State-Trait Anxiety Inventory (STAI) questionnaire, a validated six-item short-form on anxiety, which is validated for the Dutch speaking population [32].

### 2.4. Recruitment and Eligibility

Participants will be recruited from eight gynaecology and obstetrics departments in Belgium. The planned recruitment period is two and a half years. Written informed consent will be obtained at each site before any study-related activities are performed. Participants will be eligible if they fulfil the following criteria:

#### 2.4.1. Inclusion Criteria

-age 18–45 years-singleton pregnancy with ultrasound-confirmed gestational age up to eleven weeks and six days-for the group with MBS: history of SG or RYGB-participants need to speak and understand Flemish, French, or English and have e-mail access

#### 2.4.2. Exclusion Criteria

-multiple pregnancy-pregnancy beyond 12 weeks at time of inclusion-other types of MBS than SG or RYGB-known pregestational diabetes mellitus-medications known to interfere with glucose metabolism, at time of inclusion-a physical or psychological disease likely to interfere with the conduct of the study

### 2.5. Groups

#### 2.5.1. After Metabolic Bariatric Surgery

Pregnant women after SG or RYGB.

#### 2.5.2. Control

Pregnant women with no history of MBS who meet the inclusion criteria can participate if matched with a participant from the surgical group. Controls will be manually matched 1:1 based on age and pre-pregnancy BMI in early pregnancy (before twelve weeks).

### 2.6. Safety

All (serious) adverse events (S)AEs will be recorded on applicable standardised reporting forms and associated with the subject identification number, with more detailed description of the event. SAEs will be reported onwards to the sponsor, central Medical Ethics Review Committee, and national competent authorities as required.

CGM will be conducted in a masked setting utilising a blinded receiver. As such, no real-time access to glucose data will be available to participants or clinicians, and glucose values will not be used to guide clinical decisions within the study. Consequently, specific thresholds for hypoglycaemia that would prompt clinical intervention have not been defined in the protocol. Any necessary management of hypoglycaemia will fall under routine clinical care, independent of study procedures.

### 2.7. Discontinuation of Participation

Participants may withdraw consent at any time. If a participant cannot be reached for a scheduled visit, efforts will be made to contact her and any provided alternate contacts. Premature discontinuation will occur for participants experiencing adverse events or starting long-term corticosteroid treatment after inclusion, at the investigator’s discretion, as this may affect glycaemic values and interfere with CGM interpretation (intermittent prophylactic steroid use for foetal lung maturation is allowed). Noncompliance (e.g., missing study visits, or not returning CGM materials) may also lead to discontinuation, at the investigator’s discretion.

### 2.8. Study Visits and Data Collection

An overview of timing of enrolment, interventions, and assessments in the study is provided in Table 1.

#### 2.8.1. Screening Visit

The study will be introduced at first prenatal visit by a physician or midwife. Interested women will receive a summary information sheet and the informed consent form. Women will have at least 24 h to consider their decision, after which the local research team will follow up on their choice. Informed consent can only be signed if all inclusion and exclusion criteria have been met. Pregnant women can only participate in the control group if a one-on-one match is identified with a participant in the surgical group. The coordinating investigator will maintain a password-protected log containing BMI and age data for all study participants to ensure accurate matching.

#### 2.8.2. Visit 1 to 4

Women will be recruited at first prenatal visit (before 12 weeks). To evaluate glucose homeostasis, a masked CGM will be used for 10 days at the following time points: between 6 and 12 weeks of pregnancy, at 18–22 weeks of pregnancy, at 24–28 weeks of pregnancy (when screening for GDM, and at 30–34 weeks of pregnancy. At each visit with CGM, clinical examination will be performed, questionnaires will be completed, and micronutrients will be measured (as described below). As standard of care in the surgical group, follow-up by a dietician during pregnancy will be provided according to need or at the indication of the treating physician. A purpose designed multivitamin preparation (BariNutrics^®^ Prenatal) will be started at periconception. In each trimester, micronutrients will be checked by blood analyses of vitamin A, vitamin D, B12, folate, prothrombin time iron, transferrin saturation, and ferritin, for the surgical group. Deficiencies detected at any time point will be addressed by the prescription of targeted extra supplementation by the treating physician.

A standardised protocol will be used to screen for diabetes in early pregnancy and GDM later in pregnancy for both groups.

#### 2.8.3. Delivery and Early Postpartum

After delivery, umbilical cord blood will be collected for the measurement of C-peptide and stored in the biobank to allow future analyses of metabolomics, provided informed consent is present. Neonatal skinfold thickness measurement will be performed within 72 h after birth. The measurement will be performed by trained study staff using a Harpenden skinfold calliper, as previously described in the HAPO study [33]. Neonatal body fat mass will be determined according to a validated formula [25]. Maternal outcomes will be collected until maternal discharge; neonatal outcomes will be collected until neonatal discharge, which may occur jointly or separately.

#### 2.8.4. Biochemical Parameters and GDM Screening

Lab measurements will be performed in line with normal routine. Based on recommendations for micronutrient monitoring during pregnancy after MBS, vitamin A (µg/L), 25-OH-Vitamin D (µg/L), vitamin B12 (ng/L), folate (µg/L), prothrombin time (s, %, INR), iron (µg/dL), transferrin saturation (%), and ferritin (µg/L) will be measured every trimester in the surgical group [34]. Micronutrient monitoring in the control group will be conducted based on clinical judgement.

At first prenatal visit, screening for overt diabetes will be performed using fasting plasma glucose (FPG, mg/dL), or HbA1c (%) if fasting is not feasible. Women with a history of bariatric surgery will undergo screening for GDM at 24–28 weeks of pregnancy and will perform SMBG for a maximum of one week, as previously described [34]. At 24–28 weeks, the control group will be screened using a 50 g glucose challenge test. However, women with a history of GDM, obesity, or impaired fasting glycaemia in early pregnancy will undergo a 75 g OGTT, according to the 2019 Flemish guidelines [35]. An overview of diagnostic criteria for GDM in both groups is provided in Appendix A.

Specifically for this study and for all participants, at each study visit (6–12 weeks, 18–22 weeks, 24–28 weeks, and 30–34 weeks) and at delivery when cord blood is collected, two additional plasma samples will be collected for long-term storage at the biobank of UZ Leuven to allow for future analysis of metabolomics.

#### 2.8.5. Clinical Measurements

Blood pressure will be measured using a calibrated automatic blood pressure monitor. Height will be measured to the nearest 0.5 cm using a calibrated wall-mounted stadiometer. Weight will be measured to the nearest 0.1 Kg using a calibrated digital scale. BMI will be calculated as kg/m^2^. Waist circumference will be measured to the nearest 0.1 cm by applying the tape directly on the skin, horizontally at the level that is midway between the iliac crest and the lowest lateral portion of the rib cage. Body composition will be measured by bioelectrical impedance analysis with Bodystat 1500^®^ (Bodystat Ltd., Douglas, Isle of Man, UK).

#### 2.8.6. Food Diary

To assess the baseline dietary habits, a 3-day dietary record will be used, as it provides a more accurate reflection of habitual intake compared to shorter or longer periods, which are prone to misreporting. Ideally, the records will be collected on non-consecutive days [36]. For Dutch-speaking participants, the Digitaal Dagboek^®^ app will be used. This mobile application is developed by UCLL (University of Applied Sciences, Leuven, Limburg, Belgium) and is specifically designed for participants in scientific research studies. Each participant will receive a study-specific login linked to their unique subject identification number, ensuring that the study team can only access data relevant to the specific study. For the GLORIA study, participants will be able to log both prepared meals and store-bought products. The algorithm of the app will calculate daily intake of macro- and micronutrients based on their entries. For non-Dutch-speaking participants or those unable to use the app, a written food diary will serve as an alternative method of dietary data collection.

#### 2.8.7. Biobank

Participants will provide explicit consent regarding the storage of their biological material for future research. The obtained blood samples will be transported by a courier to the biobank of UZ Leuven at least every three months. Samples will be archived at the biobank UZ Leuven at −80 °C for a maximum of 10 years post-study, contingent on affirmative consent. Samples will be used in the GLORIA research project, which is fully independent of any commercial involvement. These samples either alone or in combination with the collected data can be used for further research, on the condition that this is related to the current study. Future use will require definition of the research purpose and prior ethics committee approval.

#### 2.8.8. Cord Blood

Venous cord blood (3 samples) will be collected at delivery in each participating centre by a midwife or obstetrician. Cord-blood will be obtained by free drainage (free flow) or drawn by needle aspiration (puncture) from a clamped segment of an umbilical vein. In each centre, cord blood will be stored locally at −20 °C and transported to UZ Leuven for storage at −80 °C at the biobank of UZ Leuven until analyses. Samples will be stored for max. 10 years.

#### 2.8.9. CGM

The Dexcom G6^®^ CGM system with blinded receiver (Dexcom, San Diego, CA, USA) is a CE-labelled device that is approved for use in pregnancy [37] and provides real-time continuous measurement of interstitial glucose levels via a small sensor, which is inserted in the subcutaneous skin of the abdomen or upper arm. The device is disposable and without the need for calibrations. One sensor session can record glucose values for up to 10 days. This sensor has a high accuracy, as reflected by the mean absolute relative difference (MARD) of 8.7% for wear on the posterior upper arm in pregnancy [37]. CGM will be performed at four distinct timepoints, each involving a single sensor session. Continuous CGM throughout the entire pregnancy was not pursued due to the substantial cost implications for this investigator-initiated academic trial. Additionally, continuous monitoring would likely increase participant burden, potentially affecting recruitment, adherence, and retention. The study design was therefore optimised to align with routine clinical care. Of note, previous research in uncomplicated pregnancies has demonstrated that CGM-derived metrics remain relatively stable across all three trimesters. The Dexcom G6^®^ sensor will be used in a blinded manner, so that glucose values are masked from the participant to encourage normal daily behaviour and obtain unbiased insights into maternal glucose patterns. This approach avoids potential issues associated with unblinded CGM, which may inadvertently cause anxiety, disrupt sleep, and lead to unnecessary carbohydrate intake in response to device-alerted hypoglycaemia. Such behaviours could influence participant responses and provoke post-bariatric hypoglycaemia, thereby biassing glucose data. Furthermore, the clinical significance of hypoglycaemia detected by CGM during pregnancy after MBS is not well established.

#### 2.8.10. Ultrasound Measurements

To evaluate foetal anatomy, growth, body composition, and placental function, detailed ultrasound examination according to current ISUOG guidelines will be performed by trained sonographers at approximately 12, 20, and 32 weeks of gestation (Table 1). Training of the sites will be provided by an obstetrician from UZ Leuven, who has extensive experience in ultrasound measurements in pregnant women with obesity and after bariatric surgery.

Participating centres will detach dedicated sonographers trained in routine sonographic measurements. For specific study measurements including FASTT, detailed standard operating procedures (SOPs) will be available to the research teams. Training sessions will be organised. Digital images of foetal measurements will be stored. In the data management phase, we will evaluate measurements by individual sonographer and study centre to further exclude abnormal variability (both internal and external).

#### 2.8.11. Metabolomics

The metabolomics and data analyses will be performed by the Systems Medicine Department from the Steno Diabetes centre (Copenhagen, Denmark). This lab has extensive expertise on non-targeted and targeted metabolomics and has previously shown evidence of metabolomics profile shift in diabetes complications [38]. In addition, their research group includes experts in bioinformatics to manage data from metabolomics. Cord-blood and maternal plasma samples will be collected at different time points (Table 1) and will be stored in the biobank of UZ Leuven at −80 °C before shipment to the lab of the Steno Diabetes centre. Based on a pilot study of West K et al. showing that changes in metabolites were most significant in late pregnancy [16], we will prioritise the analysis of metabolomics on maternal plasma collected at 30–34 weeks of pregnancy [16]. Since there is limited evidence that shifts in maternal metabolic profile induced by malabsorptive surgery are associated with reduced birth weight of their offspring [16], a lipidomics platform and a metabolism platform (as further described below) will be used. Based on the sample size from the pilot study (25 with RYBG and 8 SG), plasma samples from at least 60 women (30 with RYBG and 30 with SG) will be analysed and compared to at least 60 women from the control group. As metabolomics is a discovery-driven methodology, a formal power calculation was not possible due to limited existing data (the pilot study only reports changes in relative concentrations of metabolites without information on variability, and non-specific methods were used to evaluate metabolites). The SGA metabolic profile of bariatric women will be correlated as meta-analysis with publications focused on metabolic mechanisms of foetal growth retardation in populations without bariatric surgery to evaluate the impact of the surgical procedure [39]. To take into account dietary factors, the Frequency Food questionnaire [29] and data on the timing and content of the last meal before each blood sample will be collected.

Lipidomics platform: This method measures molecular lipids by ultra-high-performance liquid chromatography coupled to a quadrupole time-of-flight mass spectrometer and can be used for analysing plasma and tissue samples. Lipidomics enables the detection of approximately two thousand metabolic features and identification of 300 to 900 molecular lipids (representing the most important lipid groups, e.g., phospholipids, lysophospholipids, ceramides and sphingolipids, di- and triacylglycerols). Metabolism and Energy platform: The citric acid cycle, short-chain fatty acids, and ketones to identify up to 120 metabolites that are metabolised by both the liver and gut can be measured with gas chromatography. Then, utilising ultra-high-pressure liquid chromatography tandem mass spectrometry (UHPLC-MS-MS) near thirty-six metabolites including amino acids and related metabolites, polar metabolites, and small organic acids will be quantified; this platform is targeted and specifically designed for metabolic syndrome biomarkers. Pre-processing and statistical analysis: The data pre-processing pipelines of the metabolomic platforms are based on bioinformatic tools that automatically detect metabolite signals from the measurement mass chromatograms and associate the signals with known metabolites based on their mass-spectrometric fingerprint using in-house molecule libraries. Metabolites will be identified using semi-quantification and tandem mass spectrometry fragmentation patterns and tested for over-representation of functional annotations. Multivariate analyses will identify metabolites, ratios, and combinations of metabolites associated with clinical outcomes. Bonferroni correction will be used for multiple test correction.

### 2.9. Statistics

This study examines the association between glycaemic patterns during pregnancy after MBS and birth weight. The primary aim is to assess whether glycaemic dysregulation (e.g., hypoglycaemia, glycaemic variability) contributes to foetal growth restriction in this population.

#### 2.9.1. Sample Size and Power Calculation

The sample size (n = 95) was calculated based on the best available data on CGM in pregnancy with MBS [40,41]. The study provides 80% power to detect a moderate association (effect size = 0.3) between glycaemia measures (mean and variability) and birth weight at a significance level of *p* < 0.05. Repeated CGM measurements (four one-week periods per participant) were accounted for using a cluster size of four and assuming an interclass correlation (ICC) of 0.5. A drop-out rate of 30% was assumed, in the sense that 30% of all measurements would be missing. The sample size was decided considering adjustment for four variables in the analysis. A Bonferroni correction will be applied to control for multiple comparisons, resulting in a 2.5% test-wise significance level.

#### 2.9.2. Statistical Methods

##### Primary and Secondary Analyses (MBS Group)

The primary analysis will be performed using linear mixed models, adjusted for confounders. Mean glycemia refers to the mean interstitial glucose level measured via CGM throughout pregnancy. Glycaemic variability reflects fluctuations in blood glucose levels and is quantified using variables such as SD, CV, and MAGE derived from CGM. Repeated measurements will be used for the primary outcome analysis, with mean glycemia and glycaemic variability treated as continuous outcome variables, and birth weight as the explanatory variable. To account for clustering due to repeated measures, linear mixed models will be used with a random intercept for the subject. Confounders such as smoking, hypertension, nutrient deficiencies, insufficient gestational weight gain, and time from surgery will be added to the model as covariates to determine the independent association between birth weight and glycaemic parameters. Variables will be considered confounders if they are associated with both birth weight and glycaemic parameters. Linear slopes with 95% confidence intervals will describe the association between glycaemic parameters and birth weight. No specific action is taken to deal with missing CGM results given the fact that the analysis model is likelihood-based and known to provide unbiased results under the fairly general assumption of missingness at random [42]. Statistical analysis will be performed by SAS version 9.4; all tests will be two-sided with significance set at *p* < 0.05.

Secondary outcomes, including time spent <54 mg/dL and hyperglycaemic thresholds (>120 mg/dL and >140 mg/dL), will be analysed analogously.

Third, associations between glycaemic patterns and binary neonatal outcomes, including SGA and birth defects, will be assessed similarly using linear models for repeated measurements. Mean differences with 95% confidence intervals will be estimated between cases with and without the specified adverse outcome.

##### Exploratory Comparative Analyses

The control group will be manually matched in a 1:1 ratio based on age and pre-pregnancy BMI before participation. A comparison of both groups on participant characteristics such as parity, insufficient GWG, hypertension, micronutrient deficiencies, and smoking status will be performed using Fisher exact tests. The comparison between the MBS and control group on birth weight and glycaemic measures will be performed using linear mixed models with a random intercept to account for clustering by matching. Confounders will be included as covariates in the model in case they were shown to differ between groups. Group comparisons on binary variables such as adverse pregnancy outcomes will be performed using a similar approach, applying generalised linear mixed models for binary responses.

#### 2.9.3. Trial Management

This is an academic research trial. University Hospitals Leuven (UZ Leuven) is the sponsor and has responsibility for the overall management of the study.

#### 2.9.4. Data Management

Every site will be opened after a first initiation visit. During the study conduct, the coordinating investigator will conduct periodic quality assurance visits of all study-sites to ensure that the protocol is being followed. A first visit for all sites will be performed within three months after the first inclusion and thereafter at least once yearly. The coordinating investigator of the study may review source documents to confirm that the data recorded on CRF pages is correct. It is important that the investigator(s) and their relevant personnel are available during the quality assurance visits and audits or inspections, and that sufficient time is devoted to the process.

The collection, processing, and disclosure of personal data, such as patient health and medical information, is subject to compliance with applicable personal data protection and the processing of personal data (Regulation (EU) 2016/679 of the European parliament and of the council of 27 April 2016 on the protection of natural persons with regard to the processing of personal data and on the free movement of such data, and repealing Directive 95/46/EC (GDPR, General Data Protection Regulation)).

Each participant will receive a subject identification number to ensure confidentiality of the data. Only the local study team will have access to the file with the identifying data. The subject number will be put on the page of each visit. All data collected in this study will be referred to by subject identification number only. During the study, all data will be stored locally in each hospital at the department of obstetrics or endocrinology in a secure manner following the Belgian legislation. Participants will be asked specifically to give an informed consent for the use of their data for future studies, and for storage of biological materials. At any given time, they can ask for data and materials to be destroyed.

All amendments will be reported to the ethics committees that previously provided a favourable opinion on the study protocol.

#### 2.9.5. Datatypes Collection and Preservation of Data

All obtained data will be collected in REDCap (Research Electronic Data Capture) version 14.5.35 (Vanderbilt University, Nashville, TN, USA). Data collected in this study will be referred to by subject identification numbers only. Data will first be recorded by trained staff at each site on standardised source worksheets and thereafter in REDCap. Glucose monitoring data will initially be collected via the manufacturer’s cloud-based software using subject identification numbers. At the coordinating site, these data will be periodically exported. CGM metrics, described in Section 2.3.1, Section 2.3.2 and Section 2.3.3, will be obtained using Excel formulas and GlyCulator, a free, open-access online platform [43]. The resulting dataset will then be uploaded—per subject and per study visit—to the corresponding data instrument in REDCap using Excel-based import files. Blood samples will be stored for a max. of 10 years at the biobank of UZ Leuven to allow for additional analyses such as new biomarkers. Data will be stored during the study in each study site. Long-term storage of data lasting at least 25 years will be undertaken in collaboration with a designated data storage provider.

## 3. Discussion

This large longitudinal cohort study aims to investigate whether pregnancy following MBS is associated with altered maternal glucose profiles, as assessed by CGM, and whether these alterations are linked to differences in foetal growth and newborn body composition. Outside of pregnancy, altered glucose patterns after MBS have been well-documented [18]. Emerging data suggests that pregnancies following RYGB are marked by significant glycaemic variability, including frequent episodes of both hyper- and hypoglycaemia. However, most studies to date are small and cross-sectional. Bonis et al. first reported wide glucose fluctuations and substantial hyperglycaemia in mid-pregnancy CGM profiles of post-RYGB women, resembling patterns seen in the non-pregnant post-RYGB population [40]. Leutner et al. similarly found increased glycaemic variability, including more frequent day and nighttime hypoglycaemic events compared to pregnant controls [41]. In a larger retrospective study, Gohier et al. found that 73% of women had abnormal time in range (TIR), and found associations between hypoglycaemia and prematurity, while SGA was paradoxically linked to lower hypoglycaemia exposure [44]. Additionally, an observational cross-sectional study comparing pregnancies after RYGB and SG found that while mean glucose levels were similar, women post-RYGB experienced greater glycaemic variability and spent less time in the target glucose range, indicating more pronounced glucose fluctuations following RYGB [45]. The only longitudinal data come from Stentebjerg et al., who performed CGM across all trimesters, demonstrating that, compared to BMI- and parity-matched controls, these pregnant women spent more time in both hyperglycaemia and hypoglycaemia. They also observed a modest association between increased time spent in hypoglycaemia and SGA outcomes [21]. Overall, while evidence points to a distinct glycaemic profile in post-RYGB pregnancies, none of these earlier studies were sufficiently powered to show the implications on foetal growth [21].

This GLORIA study aims to include pregnant women with a history of both SG and RYGB, as these procedures are currently the most performed worldwide. This approach will enable us to include a representative population of pregnant women post-MBS. A second phase of the study consists of matching each participant in the cohort on a one-to-one basis with a control participant, based on age and pre-pregnancy BMI. This will allow for further, exploratory comparisons. The study will provide comprehensive data on key pregnancy outcomes, including gestational weight gain, micronutrient status, and foetal and neonatal biometry throughout pregnancy. These data, along with glycaemic patterns, will help future meta-analyses by enabling comparisons across studies using the same methods.

This study protocol has several inherent limitations. First, although the study involves multiple centres across Belgium, it only includes centres in the Flemish-speaking part. The generalizability of findings to other ethnic populations or healthcare systems may be limited. Additionally, several analyses—such as those involving metabolomics data and patient-reported outcomes—are exploratory in nature. As such, they will not provide definitive evidence but rather identify trends or associations that can inform the design of future confirmatory studies.

Nevertheless, this study is important and innovative because it will be the largest investigating glycaemic variability throughout pregnancy after MBS, measuring at four distinct time points. Our hypothesis is that it is not the woman’s weight, nor her caloric intake, but rather her (glucose) metabolism that is a key factor influencing the observed pregnancy outcomes. In other words, to improve outcomes for these pregnancies, the focus should not necessarily be on achieving appropriate gestational weight gain, but rather on preventing glycaemic variability through education and the use of novel technologies.

## Figures and Tables

**Table 1 jcm-14-04782-t001:** Schedule of enrolment, interventions, and assessments.

		STUDY PERIOD					
TIMEPOINTS in Pregnancy and Early Postpartum		Screening <12 Weeks	Study Visit 1 <12 Weeks	Study Visit 2 18–22 Weeks	Study Visit 2 24–28 Weeks	Study Visit 3 30–34 Weeks	Delivery and Early Postpartum
**ENROLMENT**	**Eligibility screening**	X					
	**Matching** control group	X					
	**Informed consent**	X					
**INTERVENTIONS**	**Masked CGM for 10 days**		X	X	X	X	
**ASSESSMENTS**	**Outcomes collected from medical records:**						
	Demographic data	X					
	Data collection from electronic medical records (medical, surgical, obstetrical)	X	X	X	X	X	X
	**Clinical and biochemical outcomes:**						
	Physical examination	X	X	X	X	X	
	BIA measurement mother		X				
	(S)AE collection		X	X	X	X	X
	foetal ultrasound		X	X		X	
	SMBG for 1 week surgical cohort				X		
	50 g GCT or 75 g OGTT control group				X		
	Skinfold measurements newborn						X
	Lab with micronutrients measurement		X	X		X	
	Lab with fasting plasma glucose or HbA1c		X				
	Metabolomics on maternal plasma		X	X	X	X	X cord blood
	**Patient-reported outcomes:**						
	Three-day food diary		X				
	Self-administered questionnaires		X	X	X	X	
	Concomitant medication	X	X	X	X	X	X

CGM: continuous glucose monitoring; BIA: bioelectrical impedance analyses; (S)AE: (serious) adverse events; SMBG: self-monitoring of blood glucose with glucometer; GCT: glucose challenge test; OGTT: oral glucose tolerance test, HbA1c: haemoglobin A1c. The X means that for a certain time point during the study, the intervention described at the left column is applicable.

## Data Availability

No new data were created or analyzed in this manuscript. Data sharing is not applicable to this article.

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
