# Peer review of "Glucose Homeostasis, Metabolomics, and Pregnancy Outcomes After Bariatric Surgery (GLORIA): Protocol for a Multicentre Prospective Cohort Study"

_jcm, 2025, doi:10.3390/jcm14134782_

Round 1

Reviewer 1 Report

Comments and Suggestions for Authors

In this protocol by Deleus and colleagues titled “Glucose Homeostasis, Metabolomics, and Pregnancy Outcomes After Bariatric Surgery (GLORIA): Protocol for a Multicentre Prospective Cohort Study”, a Belgian multicenter prospective cohort study is proposed to investigate glycaemic patterns during pregnancy in women who have undergone metabolic bariatric surgery. Women aged 18-45 years with a confirmed singleton pregnancy up to 11 weeks and 6 days and a history of Roux-en-Y gastric bypass or sleeve gastrectomy will be eligible for inclusion. The Protocol is well written and I have some suggestions for this manuscript. 

-In methods, please add information on how the biomarkers will be evaluated using Continuous Glucose Monitoring in pregnant women after Metabolic and Bariatric Surgery. 

-Define which biochemical parameters will be considered in this study.

-The supplementary material is missing.

-There are some sentences with different sizes of letters throughout the manuscript. 

-The authors should consider whether it is correct to evaluate HbA1c because HbA1c level during pregnancy is likely to be lower than the above values since HbA1c decreases during the first and second trimesters. It is necessary to clarify how gestational diabetes mellitus (GDM) will be assessed.

-In discussion section, add the main outcomes or alterations reported in the papers related to women who have undergone metabolic bariatric surgery.

Reviewer 2 Report

Comments and Suggestions for Authors

This protocol paper describes a well-designed, prospective, multicentre cohort study aiming to investigate glycemic homeostasis, metabolomics, and pregnancy outcomes in women with prior metabolic bariatric surgery (MBS), including Roux-en-Y gastric bypass and sleeve gastrectomy. The study’s strengths include its multicenter design, integration of continuous glucose monitoring (CGM), metabolomics, neonatal body composition assessments, and comprehensive clinical follow-up. However, several aspects require clarification, stronger justification, and more rigorous framing to fully establish its methodological robustness, novelty, and translational relevance.

Major Comments

  1. While the protocol is ambitious, the overall scientific rationale could be better articulated. The authors imply that glycemic variability contributes to fetal growth restriction after bariatric surgery, but a stronger mechanistic framework is needed to explain why glucose fluctuations might directly impact fetal growth beyond statistical associations.
  2. The metabolomics component is described as exploratory and discovery-driven, but currently lacks any pilot data, clear hypotheses, or detailed analytic pipelines for candidate pathway identification. Without a predefined focus (e.g., branched-chain amino acids, lipid subclasses), the clinical interpretability may be limited even if statistically significant metabolites are identified.
  3. The sample size calculation is only provided for the glycemia-birth weight association, but not for the metabolomics arm. As metabolomics will involve large-scale multiple testing, the absence of power calculations raises concern about underpowering or overfitting risk in this exploratory aim.
  4. The study uses blinded CGM measurements at four timepoints. However, glycemic variability may change dynamically between these windows, particularly around GDM screening. The protocol could better justify the choice and timing of these four measurement periods.
  5. The matching strategy for controls is limited to age and pre-pregnancy BMI, which may inadequately control for important confounders such as parity, socioeconomic status, comorbidities, or nutritional status. Matching or adjustment for these variables should be more rigorously incorporated in the analysis plan.
  6. The authors plan to use multiple statistical models (linear mixed models, generalized estimating equations, logistic regressions), but a more unified and hierarchical analytic plan would improve clarity. The primary outcome should be explicitly stated as a prespecified analysis rather than a list of exploratory endpoints.
  7. The handling of missing CGM data due to device failure or poor adherence is not adequately discussed. Imputation methods or sensitivity analyses should be pre-specified given the anticipated 30% missingness.
  8. The inclusion of multiple self-reported questionnaires adds valuable patient-reported outcomes. However, their integration into the primary scientific aims remains vague. The authors should clarify how these psychosocial variables will be used analytically (e.g., as covariates, moderators, or independent outcomes).
  9. Although multiple Belgian centers are involved, the generalizability to other ethnic populations or bariatric surgery types is limited. The authors should acknowledge this more directly in the protocol’s limitations.
  10. The mechanistic focus on maternal glycemia as the dominant factor in fetal growth is somewhat oversimplified. Alternative pathways such as placental function, inflammation, maternal gut microbiome, or epigenetic effects deserve mention to contextualize the complexity of post-bariatric pregnancy physiology.
  11. The manuscript sometimes conflates protocol-level descriptions with speculative clinical implications (e.g., interventions to reduce glycemic variability). It would be better to distinguish between planned measurements versus downstream clinical translation.
  12. The metabolomics analysis plan relies heavily on post-hoc multivariate association studies without a clearly defined causal modeling framework. The authors should clarify whether pathway enrichment, clustering, or machine learning approaches will be applied.

Minor Comments

  1. The language requires careful editing for scientific precision. Phrases such as “improve our understanding” or “uncover key metabolic components” are vague and could be replaced with more specific, hypothesis-driven objectives.
  2. The role of physical activity, as assessed by IPAQ, is mentioned but not explained in the analysis plan. Will these data be used as covariates or for stratified analyses?
  3. The duration of postpartum follow-up is unclear. Some neonatal outcomes may manifest beyond the immediate perinatal period.
  4. The description of safety monitoring is minimal. Although adverse event reporting is included, the protocol could elaborate on specific thresholds for CGM-defined severe hypoglycemia warranting clinical intervention.
  5. The rationale for using blinded CGM rather than real-time CGM could be better justified. While blinding may reduce behavioral changes, real-time monitoring could have practical benefits for pregnant women with hypoglycemia risk.
  6. Multiple self-designed questionnaires are included (e.g., for CGM usability). Some pilot validation or psychometric testing should be mentioned to support their use.
  7. The plan for biobank storage is robust, but the long-term access and governance framework for secondary data use could be better defined.
  8. While multi-omics integration is an innovative feature, the authors should explicitly state how integration across CGM, metabolomics, clinical and neonatal phenotypes will be performed analytically.

Round 2

Reviewer 2 Report

Comments and Suggestions for Authors

The authors have provided detailed and well-structured responses to the issues I raised. I believe the manuscript is now suitable for publication.